# Interactive Multi-fidelity Learning for Cost-effective Adaptation of Language Model with Sparse Human Supervision

**Jiaxin Zhang**[1], **Zhuohang Li**[2], **Kamalika Das**[1], **Sricharan Kumar**[1]
[1]Intuit AI Research    [2]Vanderbilt University
{jiaxin_zhang, kamalika_das, sricharan_kumar}@intuit.com
zhuohang.li@vanderbilt.edu

## Abstract

Large language models (LLMs) have demonstrated remarkable capabilities in various tasks. However, their suitability for domain-specific tasks, is limited due to their immense scale at deployment, susceptibility to misinformation, and more importantly, high data annotation costs. We propose a novel Interactive Multi-Fidelity Learning (IMFL) framework for the cost-effective development of small domain-specific LMs under limited annotation budgets. Our approach formulates the domain-specific fine-tuning process as a multi-fidelity learning problem, focusing on identifying the optimal acquisition strategy that balances low-fidelity automatic LLM annotations and high-fidelity human annotations to maximize model performance. We further propose an exploration-exploitation query strategy that enhances annotation diversity and informativeness, incorporating two innovative designs: 1) prompt retrieval that selects in-context examples from human-annotated samples to improve LLM annotation, and 2) variable batch size that controls the order for choosing each fidelity to facilitate knowledge distillation, ultimately enhancing annotation quality. Extensive experiments on financial and medical tasks demonstrate that IMFL achieves superior performance compared with single fidelity annotations. Given a limited budget of human annotation, IMFL significantly outperforms the $3\times$ human annotation baselines in all four tasks and achieves very close performance as $5\times$ human annotation on two of the tasks. These promising results suggest that the high human annotation costs in domain-specific tasks can be significantly reduced by employing IMFL, which utilizes fewer human annotations, supplemented with cheaper and faster LLM (e.g., GPT-3.5) annotations to achieve comparable performance.

## 1 Introduction

Large language models (LLMs) like GPT-3/ChatGPT/GPT-4 [4, 47, 5] have lately attracted great interest from both academia and industry due to their impressive in-context learning (ICL) abilities. However, the current state-of-the-art LLMs have since quickly grown from hundreds of billions [7] to even a trillion [20] parameters. Models of this scale require specialized hardware, massive-scale training data, and extensive computational power, which are inaccessible for most product or research teams. In addition, the generalizability of LLMs is predominantly decided by the scope of the underlying pre-training data. In fact, LLMs do not perform well out of the box in many real-world domains where specialized knowledge beyond the standard fields of pre-training is required (i.e., domain shifts), such as healthcare [26] and finance [46].

As an alternative to general-purpose LLMs, practitioners oftentimes find small domain-specific language models (LMs) to be more favorable as they require less training data and are faster to

37th Conference on Neural Information Processing Systems (NeurIPS 2023).

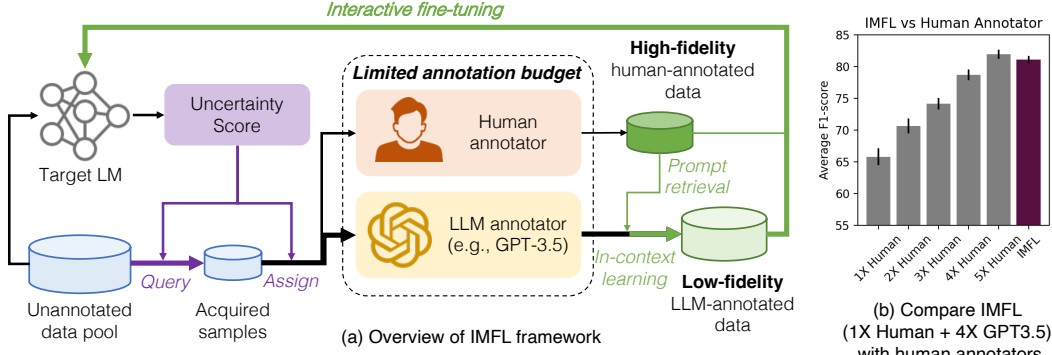

Figure 1: (a) Proposed Interactive Multi-Fidelity Learning Framework (IMFL). IMFL aims at solving the best acquisition strategy that balances between low-fidelity automatic LLM annotations and high-fidelity human annotations to maximize model performance given limited annotation budgets. (b) IMFL significantly outperforms the $3\times$ human annotation baselines in all four tasks and is very close to $5\times$ upper bound in the Headline dataset (showed). This result indicates that the high human annotation cost in domain-specific tasks can be greatly reduced by employing IMFL, which utilizes fewer human annotations combined with cheaper GPT-3.5 annotations to achieve competitive performance.

compute, leading to faster development cycles and lower operating costs [55, 12]. A common practice of developing such models is through the classic pre-training and then fine-tuning paradigm. Unfortunately, to achieve comparable performance as LLMs, tuning small LMs requires high-quality manual annotations on target domain data, which in many fields requires extensive human effort and expert knowledge, making supervised fine-tuning very expensive.

One promising approach to alleviate human annotation efforts is to leverage LLMs as knowledge bases for automatically annotating new data [43, 45]. Unfortunately, such an approach is susceptible to the misinformation [8, 39, 32, 2] of LLMs through hallucination [16, 53, 48, 30, 52], which risks generating unreliable or falsified labels and will, in turn, demolish the model's utility for high-stakes applications like healthcare and finance, where the truth is of utmost importance. As a result, the key challenge at hand is how to

Table 1: A qualitative comparison of human annotation, LLM annotation, and IMFL .

|  | Human | LLM | IMFL |
|---|---|---|---|
| Cost Saving | Low | **Very High** | **High** |
| Quality | **Very High** | Low | **High** |
| Efficiency | Low | **Very High** | **High** |
| Performance | **Very High** | Low | **High/Very High** |

effectively gather sufficient high-quality data given limited budgets on human annotation, which is a critical component in fine-tuning domain-specific LMs.

In this paper, we present a novel framework, named by IMFL, for achieving cost-effective development of domain-specific LMs, as illustrated in Fig. 1. Our approach capitalizes on the insight that different data samples inherently exhibit different levels of hardness for learning [6, 1]. Therefore, it is dispensable to request human annotation for every sample. By discerning each sample's hardness level, we can delegate the majority of the annotation tasks to automatic annotation tools such as LLMs while exclusively assigning a limited number of highly uncertain samples to human annotators, thereby reducing human effort significantly while still maintaining high annotation quality.

To improve cost efficiency, we formulate the domain-specific fine-tuning process as an *interactive multi-fidelity learning* problem. We deem LLMs and humans to be two sources of annotation with distinct fidelities, and aim to determine the optimal acquisition strategy that balances low-fidelity LLM-generated annotations and high-fidelity human annotations to maximize model performance under limited annotation budgets. We thus introduce an exploration-exploitation query strategy, wherein human annotations emphasize *exploitation* geared toward maximizing informativeness while LLM annotations concentrate on *exploration* to foster diversity and improve representatives.

To reduce the misinformation in LLM-generated annotations and improve model usability and reliability in the target domain, we incorporate two innovative designs. First, we utilize prompt retrieval to select in-context learning examples for each queried sample, thereby improving the

accuracy of LLM-generated annotations. Second, we implement variable batch sizes throughout the interactive annotation process, which manage the order in which each fidelity is chosen; this facilitates knowledge distillation and ultimately enhances annotation quality while stabilizing the LLM annotations.

We evaluate our approach on four language understanding tasks across two specialized application domains, i.e., finance and medicine. Our results highlight that LMs tuned through the proposed `IMFL` framework with GPT-3.5 as an auto-annotator significantly outperform LMs tuned with $3\times$ human annotations, and are even on par with LMs tuned with $5\times$ human annotations in some cases. In contrast to single-fidelity annotations such as only human or only LLMs, `IMFL` effectively addresses limitations related to cost saving, annotation quality, and efficiency (see Table 1). Furthermore, `IMFL` not only surpasses the performance of LLM annotators but also achieves highly competitive performance compared to human annotators, albeit at a substantially lower cost and effort.

## 2 Interactive Multi-fidelity Learning

We propose `IMFL` that builds on two key insights: (1) leveraging a substantial amount of low-fidelity annotations generated by LLMs to compensate for the insufficiency of high-fidelity human annotations during fine-tuning, and (2) utilizing high-fidelity human annotations as supervision signals to distill knowledge from LLMs while simultaneously enhancing their output annotation quality through in-context learning. Essentially, our approach, `IMFL`, can be regarded as a synergy between fine-tuning and knowledge distillation under sparse human supervision.

### 2.1 Problem Formulation

Given a total annotation budget $\mathcal{B}$ and a computational cost $\mathcal{C}$ (e.g., costs for fine-tuning, inference, and query), we aim to fine-tune a small LM $f(\boldsymbol{x}; \theta^*) : \mathcal{X} \to \mathcal{Y}$ with pre-trained parameters $\theta^*$ on a downstream task by annotating samples from an unannotated data pool $\mathcal{U} = \{x_i\}_{i=1}^U$ to constitute the annotated sample set $\mathcal{A}$ ($|\mathcal{A}| \leq \mathcal{B}$ and initially $\mathcal{A} = \varnothing$) such that its performance is maximized. Note that in our multi-fidelity setting, the annotated set contains a human-annotated subset $\mathcal{A}_H$ and an LLM-annotated subset $\mathcal{A}_G$, so $\mathcal{A} = \mathcal{A}_H \cup \mathcal{A}_G$. Similarly, the total annotation budget is composed of human annotation budget $\mathcal{B}_H$ and LLM annotation budget $\mathcal{B}_G$ ($\mathcal{B}_H$ is typically much smaller than $\mathcal{B}_G$), i.e., $\mathcal{B} = \mathcal{B}_H + \mathcal{B}_G$.

To solve for the best annotation strategy to maximize annotation and computation efficiency, we pose the annotation acquisition process as a multi-fidelity learning problem with interactions allowed for $R$ rounds. In the $r$-th round ($1 \leq r \leq R$), we query a set of instances $\mathcal{Q}^r$ and annotate acquired samples $\mathcal{A}^r$ from the unannotated pool to add annotation, i.e., $\mathcal{U} = \mathcal{U} \setminus \mathcal{A}^r$ and fine-tune the target model $f$ on $\mathcal{A}^r$ to update $\theta^{(r)}$. The goal is to minimize the empirical risk $\mathcal{R}(f)$ of the final LM $f(\boldsymbol{x}; \theta^{(R)})$ on the downstream task, subject to preset annotation budget and computational cost constraints.

### 2.2 Multi-fidelity Learning Framework

**Initialization.** We initialize the multi-fidelity learning loop by randomly selecting a small set of samples $\mathcal{A}_H^0$ from the unannotated set $\mathcal{U}$ to be annotated by human annotators. The pre-trained LM with parameters $\theta^*$ is then tuned on the initial annotated dataset:

$$\theta^{(0)} = \arg\min_{\theta^*} \frac{1}{|\mathcal{A}_H^0|} \sum_{(\boldsymbol{x}_i, y_i) \in \mathcal{A}_H^0} \mathcal{L}\left(f(\boldsymbol{x}_i; \theta^*), y_i\right), \quad i = 1, ..., n_s \tag{1}$$

where $\mathcal{L}$ is the loss function, e.g., cross-entropy for classification, and $n_s$ is the annotation size. This enables the uncertainty score of the target LM to be initially updated on domain-specific data, which helps to mitigate the *cold-start* issues [31, 49, 50].

**Interactive fine-tuning.** After model initialization, we begin query samples from the unannotated pool $\mathcal{U}^0 = \mathcal{U} \setminus \mathcal{A}_H^0$ for either human or LLM annotation. Existing methods [54] often consider the entire unannotated pool during sampling. These approaches scale poorly to large unlabeled datasets as acquiring informative samples usually involves making inferences or executing clustering which can be time-consuming if these operations were to be computed over all data samples. Thus, for any

interaction round $r$, we propose to randomly sub-sample from $\mathcal{U}^r$ to obtain a smaller candidate set $\mathcal{U}^r_s$ where the acquisition strategy can be efficiently computed.

In $r$-th round of interactive fine-tuning, we first perform the *exploration-exploitation query* (EEQ) strategy $\mathcal{S}$ (described in detail in Section 2.3) to determine the human annotation set $\mathcal{A}^r_H$ and LLM annotation set $\mathcal{A}^r_G$ from the sub-sampled unannotated pool $\mathcal{U}^r_s$. Then the interactive multi-fidelity learning can be solved by minimizing the following total loss objective:

$$\mathcal{L}_{total} = \frac{1}{|\mathcal{A}^r_H|} \sum_{(\boldsymbol{x}_i, y_i) \in \mathcal{A}^r_H} \mathcal{L}\left(f(\boldsymbol{x}_i; \theta^{(r)}), y_i\right) + \frac{1}{|\mathcal{A}^r_G|} \sum_{(\boldsymbol{x}_j, y_j) \in \mathcal{A}^r_G} \mathcal{L}\left(f(\boldsymbol{x}_j; \theta^{(r)}), y_j\right) \quad (2)$$

Unlike the existing approaches that use simultaneous annotation with equal batch sizes for each round, we emphasize the importance of annotation order (human first and then LLM) and variable batch sizes for each query step (verified in Section 4.2) and identify the following two key designs that improve query efficiency and annotation effectiveness:

*Design 1 - In-context learning with similarity-based prompt retrieval.* According to the annotation budget $\mathcal{B}^r_H$ and $\mathcal{B}^r_G$, we acquire $\mathcal{Q}^r_H$ and $\mathcal{Q}^r_G$ instances for human and LLM annotators respectively. We first annotate acquired samples $\mathcal{Q}^r_H$ by humans, obtain $\mathcal{A}^r_H$, and update the human-annotated set $\mathcal{A}_H = \mathcal{A}_H \cup \mathcal{A}^r_H$. When using LLM to automatically generate annotations for new data, we then retrieve a few examples from the current human-annotated set $\mathcal{A}_H$ as in-context examples for improving the predicted annotation quality, see Fig. 2. Leveraging recent advances in prompt retrieval [25], we compute embeddings from all annotated samples using Sentence-BERT [34] and find the most similar examples for each queried instance measured by cosine similarity. This design improves in-context learning by better utilizing human supervision which empirically helps to further improve the accuracy and robustness of LLM annotations (verified in Section 4.2). More implementation details are provided in Appendix A.

*Design 2 - Variable batch-size query.* We propose a variable batch-size query strategy that puts more human budgets towards the initial steps of the learning process to annotate the most uncertain instances and gradually decrease the batch sizes until the total budget is reached, as illustrated in Fig. 2. Naturally, another benefit of this design is that by acquiring more human-annotated examples in the early stage, we can have access to a larger pool of high-fidelity samples for conducting similarity-based prompt retrieval, which further improves the in-context learning performance and stabilizes the LLM annotations. Inspired by infinite geometric series, we design a budget decay scheme and thus set the human annotation budget for the $r$-th round to be $\mathcal{B}^r_H = \mathcal{B}_H/2^r$ and iterate until the total budget is reached, i.e.

$$\frac{\mathcal{B}_H}{2^1} + \frac{\mathcal{B}_H}{2^2} + \frac{\mathcal{B}_H}{2^3} + \frac{\mathcal{B}_H}{2^4} + \cdots + \frac{\mathcal{B}_H}{2^r} = \sum_{r=1}^{R} \left(\frac{1}{2}\right)^r \mathcal{B}_H \to \mathcal{B}_H. \quad (3)$$

Note that the residual budget after $R$ rounds will be jointly applied to the last round.

Leveraging the benefits of novel designs, we efficiently acquired larger high-quality data $\mathcal{A}^r_G$ annotated by LLMs (e.g., GPT-3.5). The next step is to update the annotated sample set in the $r$-th round $\mathcal{A}^r = \mathcal{A}^r_H \cup \mathcal{A}^r_G$ and unannotated data pool $\mathcal{U} = \mathcal{U} \setminus \mathcal{A}^r$. Then we fine-tune the target model $f$ using the annotated sample set $(\boldsymbol{x}_i, y_i) \in \mathcal{A}^r$ and update the model parameters $\theta^{(r)}$.

**Termination.** The multi-fidelity learning process is stopped if either of the two stopping criteria is satisfied: (1) Annotation budget $\mathcal{B}$: if the annotation budget after $R$ rounds is greater than the total budget limit, i.e., $\mathcal{B}_H + \mathcal{B}_G \geq \mathcal{B}$, we terminate the interactive process. (2) Computational cost $\mathcal{C}$: Compared with inference and query calculation cost, the computation cost of each fine-tuning round $\mathcal{C}_{ft}$ is typically much more expensive and we thus stop the fine-tuning process if $R \times \mathcal{C}_{ft} \geq \mathcal{C}$. Finally, we return the fine-tuned target LM $f(\boldsymbol{x}; \theta^{(r)})$ and annotated sample set $\mathcal{A}$. Algorithm 1 illustrates the step-by-step workflow of our IMFL framework.

## 2.3 Exploration-Exploitation Query Strategy

Based on the multi-fidelity learning framework, we introduce an innovative query strategy. This approach harnesses human annotation for *exploitation* by maximizing informativeness through

**Algorithm 1** IMFL framework

1: **Require**: unannotated data pool $\mathcal{U}$, target LM model $f$, query strategy $\mathcal{S}$, annotation budget $\mathcal{B}$
2: **Initialization**: $\mathcal{A} = \varnothing, \theta = \theta^{(0)}$ on $\mathcal{A}_H^0$
2: **for** rounds $r = 1, ..., R$ **do**
3:    $\mathcal{U}_s^r \leftarrow$ Extract from $\mathcal{U}$ by random sub-sampling
4:    $[\mathcal{Q}_H^r, \mathcal{Q}_G^r] \leftarrow$ Acquire $[\mathcal{B}_H^r, \mathcal{B}_G^r]$ samples by query function $\mathcal{S}$ on model $f$, data $\mathcal{U}_s^r$
5:    $\mathcal{A}_H^r \leftarrow$ Annotate acquired samples $\mathcal{Q}_H^r$ by human
6:    $\mathcal{A}_H = \mathcal{A}_H \cup \mathcal{A}_H^r$
7:    Execute prompt retrieval from $\mathcal{A}_H$
8:    $\mathcal{A}_G^r \leftarrow$ Annotate acquired samples $\mathcal{Q}_G^r$ by LLMs
9:    $\mathcal{A}^r = \mathcal{A}_H^r \cup \mathcal{A}_G^r$
10:   $\mathcal{U} = \mathcal{U} \setminus \mathcal{A}^r$
11:   $f(\boldsymbol{x}_i; \theta^{(r)}) \leftarrow$ Fine-tune $f(\boldsymbol{x}_i; \theta^{(r)})$ on $\mathcal{A}^r$
12: **return** $f(\boldsymbol{x}; \theta^{(r)}), \mathcal{A}$

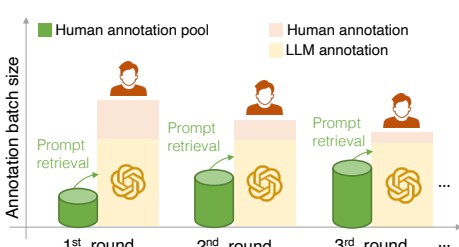

Figure 2: Illustration of interactive fine-tuning. Human annotations are first executed and the resulting annotated data are iteratively merged into the human annotation tool, which provides rich examples for prompt retrieval when calling LLM for annotation. The batch size for human annotation varies and gradually decreases as the round progresses.

uncertainty sampling, and LLM annotation for *exploration* by enhancing representativeness through diversity sampling. The core idea is a two-stage selection: executing 1) diversity sampling, e.g., selecting cluster centers to reduce intra-iteration redundancy, and 2) uncertainty sampling, e.g., selecting instances with the least confidence, to avoid inter-iteration redundancy.

Fig. 3 presents the key components and steps of the EEQ strategy. Specifically, we apply $k$-means cluster algorithm to embeddings of the sub-sampled unannotated data $\mathcal{U}_s^r$. Based on the annotation budget, we set $k = \mathcal{B}_H/2^r + \mathcal{B}_G/R$ as the clustering parameters and identify the cluster centers (or samples closest to the cluster center) as samples, thus enforcing diversity exploration. We then calculate the uncertainty score for all selected samples and rank them from high to low. The top $\mathcal{B}_H/2^r$ uncertain samples are assigned to the human annotator following the least confidence strategy:

$$\boldsymbol{x}_i^* = \arg\max_{\boldsymbol{x}_i} \left[ 1 - p(f(\boldsymbol{x}_i; \theta^{(r)}) \mid \boldsymbol{x}_i; \theta^{(r)}) \right], \tag{4}$$

which has shown to be simple and effective in a variety of settings, resulting in enforcing uncertainty exploitation [28, 42]. As discussed in Section 2.2, we then update the human-annotated pool $\mathcal{A}_H$ which enables us to retrieve a few examples as in-context examples for the LLM annotator which can annotate $\mathcal{B}_G/R$ samples with better quality and stability. More detailed discussions about the query strategy are presented in Appendix B.

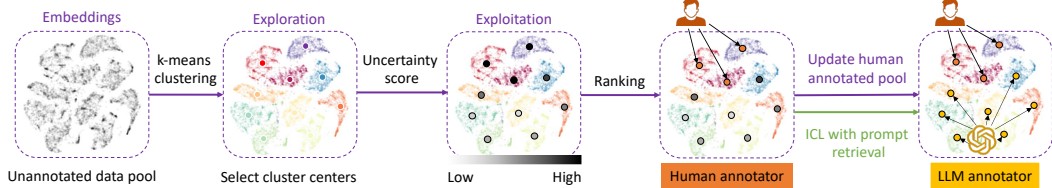

Figure 3: Illustration of exploration-exploitation query strategy with core components and steps.

# 3 Experiments

## 3.1 Datasets

We empirically validate the effectiveness of the proposed interactive multi-fidelity learning framework on four diverse datasets, spanning two important real-world application domains, namely, finance and medicine. A summary of the four datasets is provided in Table 2. For the FPB, Headline, and MedQA datasets, we use the publicly available test data for evaluation. As for the PubMedQA dataset, we follow prior work [18] and use the dev./valid data for evaluation. We evaluate the methods by (average) F-1 score for financial datasets following the same setting used by BloombergGPT [46] and accuracy for medical datasets, as is used in prior work [38, 32]. Interested readers can find more details about the datasets in Appendix D.1.

Table 2: Summary of the four domain-specific datasets used in our experiments.

| Domain | Name | Task | Size (train/test) | Metric |
|---|---|---|---|---|
| Financial | FPB [29] | Sentiment Analysis | 3876/969 | F-1 score |
| Financial | Headline [40] | News Classification | 9130/2282 | Average F-1 score |
| Medical | PubMedQA [18] | Biomedical QA | 500/500 | Accuracy |
| Medical | MedQA [17] | Medical knowledge QA | 11450/1273 | Accuracy |

## 3.2 Experiments Setup

**Fine-tuning.** We adopt Dolly 2.0, the first open-source, instruction-following LLM, as the target LM for fine-tuning. It is based on the EleutherAI Pythia [3] model family, which is a suite of decoder-only auto-regressive language models ranging from 70M to 12B parameters. Limited by our computational budget, we choose to use `dolly-v2-3b` as the pre-trained LM for our main results. We also provide additional results using larger LMs (e.g., `dolly-v2-7b` and `dolly-v2-12b`) in Appendix C.2 to show the impact of pre-trained LM size on the final performance. For efficiency, we leverage low-rank adaption techniques (LoRA) [13] to optimize the fine-tuning process for reducing memory and time cost. We execute all experiments on a GPU node with 8 NVIDIA V100 32G cores. More experiment setup details, e.g., hyperparameters, can be found in Appendix D.2.

**Query and annotation.** In the query step, we remove all labels in the training data to create a pool of unannotated data. These original ground truth labels are treated as the high-fidelity annotations provided by `Human Annotator` and are only accessed at the cost of budget consumption. For the low-fidelity annotation, we employ `GPT-3.5-turbo` as the `LLM Annotator` to automatically generate annotation for unannotated data. We note that, in reality, even collecting a large set of *unannotated* samples can oftentimes be non-trivial. As such, in our experiments, we limited our unannotated data pool to only contain 3000 data samples (randomly sampled from the original training dataset), from which we perform our query strategy. Each experiment is repeated three times and the mean is reported as the final result to reduce noise.

**Annotation and computational budget.** Unless mentioned otherwise, we assume a total annotation budget of 1000 for all datasets (see more discussions about the budget setting in Appendix C.4). As human annotation is far more expensive than using LLM (i.e., GPT-3.5) to generate annotation, we set the human annotation budget of `IMFL` to be 200 samples (20%) and the GPT-3.5 annotation budget to be 800 samples (80%). In Appendix C.5, we provide a discussion about the trade-off between annotation accuracy and cost expenses. Regarding the fine-tuning cost, we set the total number of interaction rounds for fine-tuning to be $R = 5$ to reflect the computational budgets. It is worth noting that the performance can be further improved if more rounds (i.e., a higher budget) are allowed.

## 3.3 Main results

In this section, we compare `IMFL` with single fidelity annotations to validate the effectiveness of the proposed multi-fidelity paradigm. Fig. 4 compares `IMFL` with using only human annotations, where $1\times$ Human, $3\times$ Human, and $5\times$ Human, represents the results obtained by fine-tuning on 200, 600, and 1000 human annotations, respectively. A detailed version of the main results are shown in Appendix C.1. Note that $5\times$ Human (1000 human annotations) can be seen as the performance upper bound of `IMFL` (200 Human + 800 GPT-3.5) if all budget is human annotation. From the results, we can clearly see that `IMFL` significantly outperforms the $3\times$ human annotation baselines in all four tasks. Particularly, `IMFL` achieves very close performance as $5\times$ human annotation on both Headline and PubMedQA datasets with only marginal differences (0.83% and 1.32% absolute loss respectively). This result indicates that the high human annotation cost in domain-specific tasks can be greatly reduced by employing `IMFL`, which utilizes fewer human annotations combined with cheaper and faster GPT-3.5 annotations to achieve similar performance.

Fig. 5 compares `IMFL` with using only GPT-3.5 annotations with the same total annotation budget (varied from 260 to 1000 samples). We have the following observations. First, our `IMFL` outperforms the GPT-3.5 annotation by a large margin (in terms of absolute gain) on PFB (+7.35%), Headline (+8.3%), PubMedQA (+6.89%) and MedQA (+19.95%) given the same 1000 annotation budget.

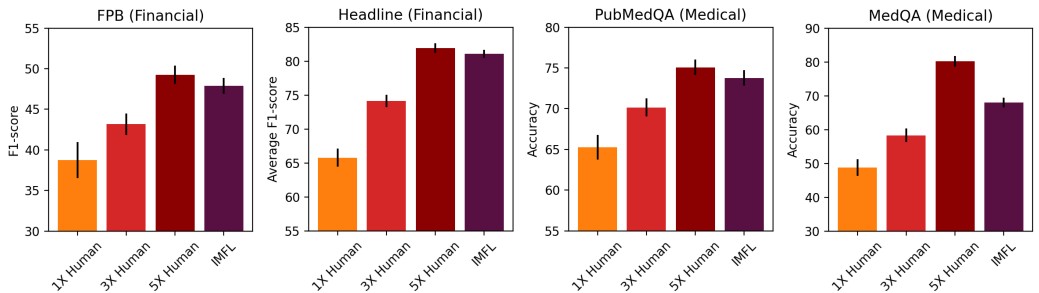

Figure 4: Comparisons between our multi-fidelity learning (200 human annotations + 800 GPT-3.5 annotations) and various sizes, i.e., 200 (1×), 600 (3×), and 1000 (5×) of human annotations.

Second, on three out of four datasets (FPB, PubMedQA, and MedQA), models tuned using IMFL with a *total* annotation budget of 260 (100 Human + 160 GPT-3.5) are able to achieve better performance than using 1000 GPT-3.5 annotations. On the Headline dataset, using 1000 GPT-3.5 annotations performs slightly better than using IMFL with a total budget of 260, but still worse if the total budget is increased to 470 ((100 + 50) Human + (160 + 160) GPT-3.5). This shows that while GPT-3.5 demonstrates promising abilities to reproduce human-generated labels [14, 56], relying solely on low-fidelity GPT-3.5 labels is not ideal for fine-tuning LMs for domain-specific tasks. In addition, compared with using only GPT-3.5 annotations, IMFL shows more reliable results with smaller variance, which benefits from a combination of human annotation and the similarity-based prompt retrieval strategy for improving the in-context learning capability of LLMs. These results verified that IMFL can efficiently utilize sparse human supervision to enhance GPT-3.5 annotations and consequently achieve better performance.

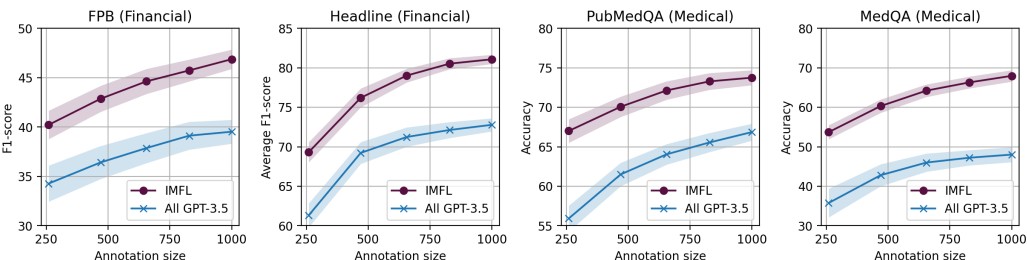

Figure 5: Comparisons between our multifidelity learning paradigm and single low-fidelity (all GPT-3.5) annotation on four domain-specific tasks given the same total 1000 annotation budget. Note that the samples for all GPT-3.5 are drawn based on the uncertainty score.

## 4 Analysis

### 4.1 Exploitation-Exploration Query vs Random Query Strategy

Table 3 compares the proposed EEQ strategy with the random query strategy on multiple settings given a limited annotation budget. Under the multi-fidelity setting, our EEQ strategy outperforms the random query strategy by a large margin (5.91% absolution gain on average). Although fine-tuning with only human annotations is supposed to produce the best results as it has the highest annotation accuracy, we observe that using only human annotations with random query is generally worse than using human and GPT-3.5 annotations with EEQ query (on three out of four datasets), thereby validating the effectiveness of the proposed EEQ query strategy. We observe that one exception is the MedQA dataset, where using only human annotations with random query performs slightly better than the proposed IMFL . This is because GPT-3.5 shows relatively low annotation accuracy on this dataset and consequently induces a negative impact on the fine-tuning performance by injecting noises into the annotated data. However, if using only GPT-3.5 annotations with the random query under the considered annotation budget, the performance would drop significantly.

Table 3: A comparison of our EEQ query strategy and random query strategy.

| Method | Budget | | Query Strategy | Dataset | | | |
|---|---|---|---|---|---|---|---|
| Multi/Single | Human | GPT-3.5 | EEQ/Random | FPB | Headline | PubMedQA | MedQA |
| Human + GPT-3.5 | 200 | 800 | EEQ | **47.88** | **81.09** | **73.76** | 67.98 |
| Human + GPT-3.5 | 200 | 800 | Random | 41.94 | 74.32 | 66.03 | 63.77 |
| Only Human | 1000 | 0 | Random | 43.81 | 75.46 | 68.87 | **70.17** |
| Only GPT-3.5 | 0 | 1000 | Random | 38.56 | 71.04 | 65.89 | 47.13 |

## 4.2 Prompt Retrieval with Variable Batch Size

To evaluate the effectiveness of similarity-based prompt retrieval and variable batch size, in Table 4, we consider several variants with random prompt retrieval and equal batch size as baselines for comparison. We find that similarity-based retrieval shows superior performance compared to the random prompt retrieval baselines (e.g., 80.28 vs 73.77 on Headline and 72.05 vs 68.10 on PubMedQA), and using variable batch size further boosts the effectiveness of retrieval by providing a larger and more diverse set of candidate examples, which is crucial in the limited budget setting.

Our interactive learning is conducted over the course of multiple rounds of interactive, with each round using a single mini-batch of data for adaptive annotation and fine-tuning. Here we also compare the interactive multiple mini-batch update strategy with the full-batch strategy, where all annotations are acquired in a single round and then used to fine-tune the model. The full-batch strategy naturally reduces the computational cost for fine-tuning but loses the benefits of interactive improvements as shown in the results. However, interestingly, we find that the similarity-based retrieval still provides a good amount of improvements in the full-batch setting which can achieve a competitive performance that is slightly better than using mini-batch updates with random retrieval. Therefore, we recommend that practitioners consider this alternative option, i.e., full-batch incorporated with similarity-retrieval, if the computational budget of fine-tuning is limited.

Table 4: Effects of prompt retrieval, variable batch size, and batch orders.

| Method | | | | Dataset | | | |
|---|---|---|---|---|---|---|---|
| Budget | Batch | Batch size | Retrieval | FPB | Headline | PubMedQA | MedQA |
| 1000 | 5 Mini-Batch | Variable | Similar | **47.88** | **81.09** | **73.76** | **67.98** |
| 1000 | 5 Mini-Batch | Equal | Similar | 46.34 | 80.28 | 72.05 | 66.11 |
| 1000 | 5 Mini-Batch | Variable | Random | 42.09 | 73.98 | 67.44 | 63.56 |
| 1000 | 5 Mini-Batch | Equal | Random | 42.34 | 73.77 | 68.10 | 63.42 |
| 1000 | 1 Full-Batch | NA | Similar | 43.72 | 75.48 | 68.90 | 63.79 |
| 1000 | 1 Full-Batch | NA | Random | 39.80 | 72.11 | 65.94 | 57.23 |

## 4.3 Alternative Query Strategies

Besides the random query and the proposed EEQ query, we also explore several additional query strategies for interactive multi-fidelity learning, including (i) confidence-based strategies: predictive entropy (Entropy) [35], least confidence (Least-c) [22], breaking ties (Breaking-t) [27]; (ii) diversity-based strategies: K-means [50], Diversity [37]; and (iii) hybrid strategy [19], a combination of confidence and diversity by a weighted sum. As shown in Table 5, our EEQ strategy outperforms all the other methods on two representative tasks (Headline and MedQA). These alternative strategies are simple to implement and perform better than the random baseline. Unfortunately, directly applying them to our multi-fidelity paradigm does yield the most desirable performance. The one that achieves the closest performance is the hybrid strategy. However, it ignores the effects of annotation orders and fidelity which are important ingredients for achieving high performance in our multi-fidelity setting.

## 4.4 Annotation Accuracy by Different GPT-based Annotators

The fine-tuning performance relies on the annotation accuracy of the LLM annotator, since noisy annotations may hurt the final model performance in terms of accuracy and reliability. Here we focus

Table 5: A comparison of various alternative query strategies on two representative tasks.

| Dataset | Random | Entropy | Least-c | Breaking-t | K-means | Diversity | Hybrid | EEQ |
|---------|--------|---------|---------|------------|---------|-----------|--------|-----|
| Headline | 74.32 | 76.42 | 77.55 | 77.34 | 76.59 | 77.61 | 79.23 | **81.09** |
| MedQA | 63.77 | 65.15 | 65.21 | 65.28 | 66.44 | 66.41 | 66.94 | **67.98** |

on evaluating the annotation accuracy of different variants of GPT (i.e., GPT-3 vs GPT-3.5) instead of fine-tuning accuracy through multiple experiments. We notice that GPT-3.5 annotation with zero shot performs much worse than few-shot and our retrieval methods in domain-specific tasks. This is because although naive GPT-3.5 shows promising performance in doing zero/few-shot learning in-distribution scenarios, it lacks domain knowledge to make accurate predictions in the considered out-of-domain tasks. The proposed prompt retrieval leveraging human annotations from domain experts with in-context learning capabilities of LLMs substantially improves the performance of GPT-3.5 annotation.

Table 6: A comparison of annotation accuracy by GPT-3 and GPT-3.5 in zero/few-shot learning.

| Dataset | GPT-3 Annotation | | | GPT-3.5 Annotation | | |
|---------|-----------|--------|--------|-----------|--------|--------|
| | retrieval | 5-shot | 0-shot | retrieval | 5-shot | 0-shot |
| Headline | 75.59 | 72.51 | 70.25 | **79.40** | 76.15 | 73.31 |
| MedQA | 51.42 | 44.89 | 42.03 | **59.45** | 53.57 | 50.82 |

If the annotation budget is very limited, GPT-3 is a cheaper alternative but underperforms GPT-3.5 even with prompt retrieval applied. In contrast, we have a chance to utilize GPT-4 (more expensive and limited access) for annotation in our multi-fidelity paradigm. Please see additional GPT-4 annotation results in Appendix C.3. Recent work shows promising capabilities of GPT-4 on medical challenge problems. For example, GPT-4 achieves 81.38 (5-shot) and 78.87 (0-shot) for the MedQA task as reported by [32]. Note that `IMFL` uses GPT-3.5 as our LLM annotator but is easy to extend to other LLM annotators, e.g., GPT-based models or open-source models such as LLaMA which depends on the annotation budget. An exhaustive study of different LLM annotators is beyond the scope of this work.

### 4.5 Ablation Study of Human Annotation Ratio

Given a total 1000 annotation budget, a key question is how to assign the budget to human annotators or GPT-3.5 annotators. In our original setting, we use a 20/80 ratio since the human annotations are much more expensive (money cost, time cost, and training cost, specifically in domain-specific areas, e.g., finance and medicine) than GPT-3.5 annotations. We also expect to minimize the ratio of human annotations so we conduct an ablation study for 10/90 and 5/95 ratios and evaluate their effect on performance in our framework. Table 7 shows the performance comparisons of various ratios of human annotations, i.e., $0.5\times$ and $0.25\times$ human annotations. We can note that the performance drops obviously when less human effort is conducted. For the case of $0.5\times$ human annotations, it is lower than our original setting but still comparable to $3\times$ human annotations. However, the case of $0.25\times$ human annotations shows a significant decrease because too few human annotations weaken the effect of in-context prompt retrieval and reduce the accuracy of initial uncertainty estimation. In short, a certain amount of human annotation is necessary for our framework even though we seek minimal human effort. We thus need to consider a trade-off between accuracy and annotation budgets.

Table 7: Performance comparisons of various ratios of human annotations on four datasets.

| Method | Number of Annotations | | Dataset | | | |
|--------|---------|---------|---------|----------|----------|-------|
| | Human | GPT-3.5 | FPB | Headline | PubMedQA | MedQA |
| IMFL | 200 (1×) | 800 | **47.88 ± 0.98** | **81.09 ± 0.58** | **73.76 ± 0.95** | **67.98 ± 1.45** |
| IMFL | 100 (0.5×) | 900 | 43.66 ± 1.42 | 75.41 ± 1.01 | 70.88 ± 1.08 | 61.44 ± 1.83 |
| IMFL | 50 (0.25×) | 950 | 40.76 ± 1.48 | 73.65 ± 1.09 | 68.18 ± 1.11 | 52.38 ± 1.93 |

## 5 Related Work

**Domain-specific LLMs.** The significance of domain-specific training for encoder-only masked language models is widely recognized. Two commonly adopted methods are to either train BERT models [9] from scratch using domain-specific data or to continue pre-training an existing model on new domain-specific data. Following these approaches, several BERT-based models are built by domain experts, e.g., BioBERT [21], ClinicalBERT [15], etc. A recent trend is to train decoder-only models by utilizing domain-specific data, such as Med-PaLM [38], BioGPT [26] and BloombergGPT [46]. These findings highlight the advantages of in-domain pre-training, especially if sufficient data is available. However, an underlying challenge is how to train domain-specific LMs when there is insufficient data for large-scale pre-training due to a limited annotation budget. Our work addresses this underexplored problem by developing a cost-effective fine-tuning paradigm with limited budgets of human annotation.

**Multi-fidelity learning.** Our work is motivated by recent findings that suggest GPT models are capable of replicating or even outperforming human annotation as indicated by several studies [14, 43, 56]. However, these approaches suffer from unreliable annotations and lack confidence when applied to domain-specific tasks, especially in specialized high-stakes fields like finance and medicine. Our key idea originates from multi-fidelity optimization approaches [33, 23, 24] that optimize the objective function by utilizing varying approximations with different levels of precision and cost. Previous studies in the field of NLP have explored "dual supervision" to train models by combining two types of labels [45]. In contrast to this naive combination approach, we delve into a novel multi-fidelity framework that achieves cost-effective adaptation of domain-specific language models through fine-tuning and in-context learning.

**Active learning.** Active Learning (AL) is an extensively studied field concerned with improving the performance of language models with fewer labeled instances [54, 44, 31]. Current works within AL typically focus on two main scenarios: active fine-tuning [42, 28] and active in-context learning [10, 41]. The former involves iteratively updating model parameters but is not well-suited for directing training/fine-tuning LLMs such as GPT-3.5 which would induce high computational costs. Conversely, the latter is efficient but the performance solely relies on the few-shot learning ability of LLMs, which is unreliable for domain-specific tasks that require expert knowledge beyond standard pre-training data. In contrast, the proposed IMFL, which fully utilizes few high-fidelity annotations from human annotators to guide the LLM-annotator, can be regarded as synergizing the power of both fine-tuning and knowledge distillation from LLMs under sparse human supervision. Our experiments demonstrate that our approach can significantly reduce human annotation efforts while achieving highly competitive performance given a limited budget for annotation and computational resources, which enables flexible and effective deployment in real-world applications.

## 6 Discussion and Limitation

We compare IMFL to single fidelity annotations to evaluate the effectiveness of our proposed multi-fidelity paradigm. The extensive experimental results reveal that employing IMFL can significantly reduce the high cost of human annotation in domain-specific tasks. Furthermore, we demonstrate that IMFL efficiently uses sparse human supervision to improve GPT-3.5 annotations through prompt retrieval and in-context learning, ultimately leading to enhanced performance.

Despite the promising performance, we note there are certain *limitations* to our approach. First, the current IMFL framework assumes that the annotation budget is defined by the number of annotations, rather than reflecting the true cost which typically involves multiple complex factors (e.g., administrative cost, training cost of human annotators, time, etc.) in real-world scenarios. Second, IMFL's performance is limited by the size of the unannotated dataset and the diversity of examples presented in the dataset as IMFL only seeks to improve performance through annotating existing samples rather than creating new samples. Lastly, limited by budgets and the capacity of the LM to be fine-tuned, IMFL does not achieve state-of-the-art performance in some general NLP tasks, where directly adopting the latest LLMs remains a better choice. Nevertheless, we anticipate the performance of IMFL to continue to grow by incorporating stronger LLM annotators, such as GPT-4, to further improve annotation accuracy. We leave this as our future work.

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
