# Appendix

## A   Prompt Retrieval

### A.1   Implementation Details

The goal of prompt retrieval is to retrieve annotated data instances that are semantically similar to the query as in-context learning examples to improve the LLM annotator's performance without the need to fine-tune it. Intuitively, the selected instance-annotation pairs would serve as an illustration to facilitate generating better and more accurate annotations for domain-specific applications.

To operationalize this idea, we follow existing studies to retrieve annotated instances that are close to the queried instance in some embedding space that captures semantic or lexical similarities. The design of prompt retrieval is originally described in Section 2.2. Here we provide a more detailed description of the procedure below. Given the queried instance $x$, annotated data pool $\mathcal{A}$, sentence encoder model (Sentence-BERT [34]), and the number of in-context examples $k$, we execute the following steps:

1. Compute the embeddings of the queried instance and instances from the annotated pool.

2. We retrieve the nearest $k$ neighbors of the queried instance $x$, denoted as $x_1, x_2, ..., x_k$, from $\mathcal{A}$ according to a pre-defined similarity metric (e.g., pair-wise cosine distance) measured in the embedding space.

3. The selected neighboring annotated instances are sorted from high similarity to low similarity.

4. The $k$ instances are concatenated with their annotations to form the in-context examples, which are then used to construct the prompt for the queried instance to generate annotation from the LLM annotator.

### A.2   Embedding Models

Our proposed framework is flexible with different embedding models. We choose to use Sentence-BERT in our implementation because it is a general-purpose sentence encoding model that can be employed in different tasks from diverse domains without requiring task-specific fine-tuning. Although encoders fined-tuned on domain-specific tasks would potentially further improve the performance by making the prompt retrieval more effective, it would require more in-domain annotated data, which conflicts with our objective of designing a cost-effective adaptation framework.

### A.3   Hyperparameters

We have summarized additional hyperparameters related to prompt retrieval in Table 8.

Table 8: Hyperparameters used in prompt retrieval

| Hyperparameters | Values |
|---|---|
| Sentence encoder model | sentence-transformers/all-mpnet-base-v2 |
| Similarity metric | cosine similarity |
| Pooling Method | mean pooling |
| Dimension of embeddings | 768 |
| Number of clusters | annotation budget at a specific iteration |
| Number of neighbors | 50 |
| Number of retrieved examples | 5 |
| Size of unlabeled data pool | 3000 |

# B  Exploration-Exploitation Query Strategy

## B.1  Uncertainty Score

For the tasks of natural language generation, the probability of the entire sequence, $\boldsymbol{s}$, is the product of the conditional probabilities of new (next) tokens given past tokens, whose resulting log-probability is $\log p(\boldsymbol{s}|x) = \sum_{i=1}^{n} \log p(s_i|\boldsymbol{s}_{<i})$, where $s_i$ is the $i$-th output token and $\boldsymbol{s}_{<i}$ denotes the set of previous tokens. In this work, we define the uncertainty score $C$ by using the geometric mean of the token-level log-probabilities: $C = \frac{1}{n} \sum_{i=1}^{n} \log p(s_i|\boldsymbol{s}_{<i})$. Since the target LM is an offline model with complete white-box access to its token-level log-probability, the uncertainty score can be easily calculated during model inference.

## B.2  Redundancy Reduction

The proposed EEQ strategy aims to reduce both intra-iteration and inter-iteration redundancy. Typically, uncertainty-based methods acquire similar samples within an iteration, known as intra-iteration redundancy, while diversity-based approaches acquire similar samples across iterations, known as inter-iteration redundancy. Existing hybrid methods try to avoid intra- and inter-iteration redundancies by combining diversity and uncertainty sampling [51]. Still, they may suffer from these redundancies due to unifying the uncertainty and diversity objectives into a single query function, which tends to prioritize one objective over the other.

Our EEQ strategy is an independent two-step selection by first executing the diversity sampling and then the uncertainty sampling. In the first step, we select a subset of an unlabeled data pool consisting of diverse data points in the embedding space. In the second step, we acquire high-uncertainty data points (the ones that are predicted with low confidence by the current model) from the subset.

# C  Additional Experiment Results

## C.1  Detailed Main Results

We provide a detailed version of our main results in Figure 4 and Figure 5 (main paper), where the mean value and standard deviation among the three trials are reported. We also add more comparisons including 2× human annotations and 4× human annotations in Table 9.

Table 9: Detailed main results about comparisons between our methods on various cases.

| Method | Number of Annotation | | Dataset | | | |
|---|---|---|---|---|---|---|
| | Human | GPT-3.5 | FPB | Headline | PubMedQA | MedQA |
| 1× Human | 200 | 0 | $38.74 \pm 2.23$ | $65.76 \pm 1.33$ | $65.28 \pm 1.51$ | $48.77 \pm 2.42$ |
| 2× Human | 400 | 0 | $40.23 \pm 1.99$ | $69.77 \pm 1.16$ | $67.64 \pm 1.18$ | $52.53 \pm 2.00$ |
| 3× Human | 600 | 0 | $43.16 \pm 1.31$ | $74.13 \pm 0.91$ | $70.15 \pm 1.11$ | $58.31 \pm 1.98$ |
| 4× Human | 800 | 0 | $47.32 \pm 1.30$ | $77.59 \pm 0.85$ | $72.66 \pm 0.83$ | $72.49 \pm 1.67$ |
| 5× Human | 1000 | 0 | $49.24 \pm 1.16$ | $81.92 \pm 0.71$ | $75.08 \pm 0.97$ | $80.23 \pm 1.57$ |
| All GPT-3.5 | 0 | 1000 | $39.53 \pm 1.21$ | $72.79 \pm 0.83$ | $66.87 \pm 1.07$ | $48.03 \pm 1.90$ |
| IMFL (ours) | 200 | 800 | $47.88 \pm 0.98$ | $81.09 \pm 0.58$ | $73.76 \pm 0.95$ | $67.98 \pm 1.45$ |

## C.2  Effect of Model Size

Fig. 6 shows performance with varying sizes of language models, i.e., `dolly-v2-3b` [1], `dolly-v2-7b` [2] and `dolly-v2-12b` [3] on FPB, Headline, PubMedQA, and MedQA datasets. In general, the performance is improved as the model size increases but the range of improvement is not significant. The performance gap between `dolly-v2-3b` and `dolly-v2-7b` is slightly larger than the gap

---

[1] https://huggingface.co/databricks/dolly-v2-3b
[2] https://huggingface.co/databricks/dolly-v2-7b
[3] https://huggingface.co/databricks/dolly-v2-12b

between `dolly-v2-7b` and `dolly-v2-12b`. The minimum/average/maximum increment from 3b to 12b is 1.66 (PubMedQA), 2.72, and 5.23 (MedQA).

As Databricks claimed, `dolly-v2-12b` is not a state-of-the-art model but does exhibit surprisingly high-quality instruction following behavior not characteristic of the foundation model on which it is based. Our target is to build a cost-effective multi-fidelity learning framework that allows us to develop smaller and faster domain-specific models for real-world production. As a result, we prefer to use a smaller version of the based model which shows competitive performance as the larger one but significantly reduces the computational costs, e.g., fine-tuning, inference, and hardware requirements.

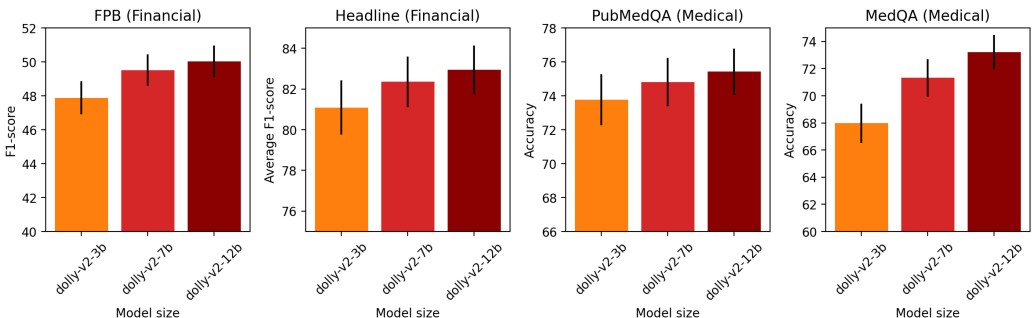

Figure 6: Comparisons of various models (`dolly-v2-3b`, `dolly-v2-7b` and `dolly-v2-13b`) with our multi-fidelity learning (200 human annotations + 800 GPT-3.5 annotations).

### C.3 Effect of annotators (GPT-3 vs GPT-3.5 vs GPT-4)

Our framework is flexible with any LMs for annotations. We have investigated the impact of different models by comparing GPT-3.5 with GPT-3 (see Table 6), as GPT-4's API was not available at the time of writing. Our observation is that GPT-3.5 provides better annotation quality but at the cost of a (slightly) higher price. Per the reviewer's request, we have added a comparison of the annotation quality of GPT-3.5 and GPT-4 and summarized the results in the following tables. GPT-4 annotator shows better accuracy but the price is much higher than GPT-3.5 and GPT-3. We will add them to the updated version.

Table 10: A comparison of annotation accuracy by GPT-3, GPT-3.5, and GPT-4 in zero/few-shot learning.

|  | GPT-3 Annotation | | | GPT-3.5 Annotation | | | GPT-4 Annotation | | |
| --- | --- | --- | --- | --- | --- | --- | --- | --- | --- |
|  | retrieval | 5-shot | 0-shot | retrieval | 5-shot | 0-shot | retrieval | 5-shot | 0-shot |
| Headline | 75.59 | 72.51 | 70.25 | 79.40 | 76.15 | 73.31 | 80.13 | 78.34 | 77.20 |
| MedQA | 51.42 | 44.89 | 42.03 | 59.45 | 53.57 | 50.82 | 82.67 | 81.38 | 78.87 |

### C.4 Budget setting

In our experiments, we select a default annotation budget of 1000 based on the following considerations: (1) Computational Costs: Using a larger annotation budget would result in an increasing amount of annotation and computational costs for both our experiments and practical deployment. An annotated budget of $\leq 1000$ is a typical setting used by many prior works [11, 50, 36, 28] that consider the low-resource setting. (2) Dataset Annotation Constraints: The human annotation budget for our experiments is also limited by the availability of annotated domain-specific datasets. For example, PubMedQA only has 1000 expert-annotated QA instances [18]. In general, the authors believe an annotation budget of 1000 is practically meaningful for NLP tasks in specialized domains like medicine and finance. Such experimental design choices should not affect the validity of our findings.

## C.5 Discussions of Trade-off between Quality and Cost

It is essential to determine the desired level of annotation quality for the specific task at hand. Our results suggest that although more expensive options like GPT-4 often show higher annotation accuracy, the performance gap between it and other cheaper options, e.g., GPT-3 or GPT-3.5, varies across different tasks. For instance, on the Headline dataset, GPT-3.5 offers a similar annotation accuracy as GPT-4 (79.4 vs 80.13) while being 20x cheaper ($0.0015 per 1k tokens vs $0.03 per 1k tokens), which makes it the better choice on this task. Additionally, the selection of LM annotation depends on budget constraints. If given a very limited budget, the expensive option may not be a valid option as it cannot provide a sufficient quantity of annotations. In practice, we recommend running a small-scale pilot study on the task at hand to compare the trade-offs of different LM annotations by defining some simple proxy metrics before performing large-scale annotation and fine-tuning. In general, it is challenging to choose the optimal annotation LM without defining some quantitive measures of the "value for money", which, as mentioned in our discussion section, is still an open research problem since modeling and optimizing such cost in real-world settings involves many complex factors, e.g., task complexity, desired label quality, and available resources.

# D  Additional Experiment Details

## D.1  Dataset Details

Four datasets from financial and medical domains are introduced below. For each data, we use the standard train/test split available from the HuggingFace/Github sources. For each dataset, we use the test data (available publicly) for evaluation (FPB, Headline, and MedQA). Otherwise, we follow prior work [18] and use the dev./valid data for evaluation (PubMedQA). We evaluate the methods by (average) F-1 score for financial datasets following the exact setting of BloombergGPT [46] and accuracy for medical datasets, exactly matching the metrics used in [38, 32].

- **FPB** (Financial Domain) [4]: The Financial Phrasebank Dataset [29] includes a sentiment classification task on sentences from financial news. Any news that could benefit/hurt an investor is considered positive/negative and neutral otherwise. We report the F-1 score as the evaluation metric.

- **Headline** (Financial Domain) [5]: The News Headline Classification [40] is a binary classification task for predicting whether a news headline in the gold commodity domain includes certain information. This human-annotated dataset contains English news headlines about "gold". Each news article carries a subset of the 9 tags: "price or not", "price up", "price down", "price stable", "past price", "future price", "past general", "future general", "asset comparison". We report the average weighted F1 score across all categories.

- **MedQA** (Medical Domain) [6]: The MedQA dataset [17] consists of US Medical License Exam (USMLE) style questions. These questions are obtained with a choice of 4 or 5 possible answers from the National Medical Board Examination in the USA, which are used to evaluate human doctors' professional knowledge and ability to make clinical decisions. These exams cover various questions and generally require a deep understanding of related medical concepts learned from medical textbooks to answer. We consider the 4-option version in our experiment.

- **PubMedQA** (Medical Domain) [7]: PubMedQA [18] is a biomedical QA dataset collected from PubMed abstracts. The task of PubMedQA is to answer research questions with yes/no/maybe provided with the corresponding abstracts. PubMedQA has 1k expert-annotated, 61.2k unlabeled, and 211.3k artificially generated QA instances.

We have summarized the additional statistics in Table 11, with the average, minimum, and maximum sentence length (in tokens):

---

[4] https://huggingface.co/datasets/financial_phrasebank
[5] https://www.kaggle.com/datasets/daittan/gold-commodity-news-and-dimensions
[6] https://github.com/jind11/MedQA
[7] https://huggingface.co/datasets/pubmed_qa

Table 11: Statistics of datasets: average, minimum, and maximum sentence length.

| Sentence length | FPB | Headline | MedQA | PubMedQA |
|---|---|---|---|---|
| Average | 28.16 | 12.98 | 167.29 | 20 |
| Minimum | 2 | 2 | 17 | 6 |
| Maximum | 148 | 39 | 551 | 50 |

## D.2 Experiment Setup Details

**Variable batch size** According to our budget decay scheme, the total budget (200 human annotations + 800 GPT-3.5 annotations) allocated for human annotation is 100, 50, 25, 13 (rounded up from 12.5), and 12 (rounded up from 6.5 plus 5 in residual), for rounds 1 to 5 respectively. In contrast, the budget allocated for GPT-3.5 annotation remains constant at 160 for each round. Therefore, the combined budget for both human and GPT3.5 annotation gradually increases to reach a cumulative budget of [260, 470, 655, 828, 1000] from round 1 to 5.

**Fine-tuning details** Our fine-tuning hyperparameters are listed in Table 12. Given the following hyperparameter settings, LoRA [13] can fine-tune the dolly models on 8 V100 GPUs with 32 GB RAM using bfloat-16 precision training. For our `dolly-v2-3b`, one fine-tuning round can be finished in 10-15 minutes.

Table 12: Details of hyperparameters in fine-tuning process

| Hyperparameters | Values |
|---|---|
| learning_rate | 2E-5 |
| batch_size | 64 |
| micro_batch_size | 4 |
| gradient_accumulation_steps | batch_size // micro_batch_size |
| num_epochs | 3 |
| weight_decay | 0.0 |
| lora_r | 4 |
| lora_alpha | 16 |
| lora_dropout | 0.05 |
| warmup_steps | 100 |