# OpenReview forum: "Interactive Multi-fidelity Learning for Cost-effective Adaptation of Language Model with Sparse Human Supervision"
_NeurIPS.cc/2023/Conference — NeurIPS 2023 poster_

### Official Review · Reviewer_ogFx · 2023-06-30

**Soundness:** 3 good
**Presentation:** 3 good
**Contribution:** 3 good
**Rating:** 6
**Confidence:** 4

**Summary:**

The paper proposes the Interactive Multi-Fidelity Learning (IMFL) framework to develop small domain-specific Large Language Models (LLMs) under limited annotation budgets. Specifically, IMFL balances low-fidelity LLM annotations and high-fidelity human annotations to maximize model performance. Experiments on four domain-specific tasks demonstrate that IMFL outperforms single fidelity annotations, offering a cost-effective solution for domain-specific LLM development.

**Strengths:**

1. The paper addresses the practical challenges of deploying large language models in domain-specific tasks, such as their scale and high annotation costs.

2.  This work considers the practical setting of addressing the challenges of deploying LLMs, hence is somehow realistic.

3. The IMFL framework offers an effective solution for the cost-effective development of small domain-specific LLMs by leveraging multi-fidelity learning.

4. Experiments on financial and medical tasks provide empirical evidence of the superiority of IMFL over single fidelity annotations.

**Weaknesses:**

1. While the paper demonstrates superior performance compared to single fidelity annotations, it would be valuable to provide a more comprehensive comparison with alternative approaches or baselines to highlight the specific advantages of IMFL.

2. It would be interesting if the author can provide more rationale for the selection of the dataset.

**Questions:**

1. Have you considered studying the effect of using GPT-3.5 and GPT-4 as annotators in your research? It would be interesting to explore how the performance and annotation quality differ when employing these different versions of the language model (with different prices/label quality) as annotators.

2. Have you tried on other AL baselines such as [14-16] mentioned in your paper? This would make the results with more AL baselines more insightful.

**Limitations:**

Yes

---

> ### Author Rebuttal · Authors · 2023-08-09
>
> Dear reviewer ogFx,
>
> Thank you very much for taking the time to engage with our paper thoroughly, and constructive comments. We’re happy to hear that you thought our work is practical and effective, addressing the ground challenges of deploying LLMs in domain-specific tasks. Please find our response below:
>
> > **It would be interesting if the author can provide more rationale for the selection of the dataset.**
>
> Thanks for the great suggestion. Our focus in this work is to build a novel end-to-end framework for cost-effective fine-tuning of LMs on specialized domain-specific tasks that are beyond the scope of standard pre-training. Thus for evaluation, we mainly select datasets that focus on domain-specific applications with direct real-world implications, such as finance and medicine.  The finance datasets are selected following the BloombergGPT [1] paper. The medical datasets are selected following prior work such as MedPaLM [2].  To increase the diversity of datasets and tasks, we select FPB to cover sentiment analysis and Headline to cover news classification, while PubMedQA and MedQA are very commonly used QA datasets for evaluating the capabilities of LMs in the medical domain.
>
>
> [1] Wu, Shijie, Ozan Irsoy, Steven Lu, Vadim Dabravolski, Mark Dredze, Sebastian Gehrmann, Prabhanjan Kambadur, David Rosenberg, and Gideon Mann. "Bloomberggpt: A large language model for finance." arXiv preprint arXiv:2303.17564 (2023).
>
> [2] Singhal, Karan, Shekoofeh Azizi, Tao Tu, S. Sara Mahdavi, Jason Wei, Hyung Won Chung, Nathan Scales et al. "Large language models encode clinical knowledge." Nature (2023): 1-9.
>
> > **Have you considered studying the effect of using GPT-3.5 and GPT-4 as annotators in your research? It would be interesting to explore how the performance and annotation quality differ when employing these different versions of the language model (with different prices/label quality) as annotators.**
>
> Thank you for the comment. Our framework is flexible with any LMs for annotations. We have investigated the impact of different models by comparing GPT-3.5 with GPT-3 (see Table 6), as GPT-4’s API was not available at the time of writing. Our observation is that GPT-3.5 provides better annotation quality but at the cost of a (slightly) higher price. Per the reviewer’s request, we have added a comparison of the annotation quality of GPT-3.5 and GPT-4 and summarized the results in the following tables.  GPT-4 annotator shows better accuracy but the price is much higher than GPT-3.5 and GPT-3.  We will add them to the updated version.
>
> |          | GPT-3 Annotation |        |        | GPT-3.5 Annotation |        |        | GPT-4 Annotation |        |        |
> |:--------:|:----------------:|:------:|:------:|:------------------:|:------:|:------:|:----------------:|:------:|:------:|
> |          |     retrieval    | 5 shot | 0 shot |      retrieval     | 5 shot | 0 shot |     retrieval    | 5 shot | 0 shot |
> | Headline |       75.59      |  72.51 |  70.25 |        79.4        |  76.15 |  73.31 |       80.13      |  78.34 |  77.2  |
> |   MedQA  |       51.42      |  44.89 |  42.03 |        59.45       |  53.57 |  50.82 |       82.67      |  81.38 |  78.87 |
>
> > **Have you tried on other AL baselines such as [14-16] mentioned in your paper? This would make the results with more AL baselines more insightful.**
>
> Our work focuses on addressing the problem of cost-effective adaptation of LMs for domain-specific applications where existing works [14-16] are not directly applicable. Specifically, [14] studies contrastive AL and is computationally intensive, as its effectiveness relies on using a large batch size, making it unsuitable for low-resource settings. [15] and [16] mainly focus on the cold-start problem in AL which is orthogonal to our work. Advances in this line of research might be helpful to improve our framework in the cold-start scenarios, but a nuanced study is beyond our focus.

---

> > ### Comment · Reviewer_ogFx · 2023-08-12
> > **Thanks for your rebuttal!**
> >
> > Thanks for your response.
> >
> > For the question of "studying the effect of using GPT-3.5 and GPT-4 as annotators", I am actually curious about the authors' thoughts on the **optimal selection of the labeling sources**. I feel the findings of the "GPT-4 annotator shows better accuracy but the price is much higher than GPT-3.5 and GPT-3." is interesting, and I am also interested in a question as **"if the annotation budget (in $) is given, which labeler should we use? Should we use cheaper annotators with lower quality, or using more expensive ones to create a smaller but more precise labeled set?"** I think it would be even better if the authors could provide more insights on this.

---

> > > ### Author Response · Authors · 2023-08-12
> > > **Response to reviewer ogFx**
> > >
> > > We appreciate the reviewer's insightful questions. Our study aimed to shed light on these important considerations, and we are glad to provide additional insights.
> > >
> > > It is essential to determine the desired level of annotation quality for the specific task at hand. Our results suggest that although more expensive options like GPT-4 often show higher annotation accuracy, the performance gap between it and other cheaper options, e.g., GPT-3 or GPT-3.5, varies across different tasks. For instance, on the Headline dataset, GPT-3.5 offers a similar annotation accuracy as GPT-4 (79.4% vs 80.13%) while being 20x cheaper (0.0015 dollar 1k tokens vs 0.03 dollar 1k tokens), which makes it the better choice on this task.  Additionally, the selection of LM annotation depends on budget constraints. If given a very limited budget, the expensive option may not be a valid option as it cannot provide a sufficient quantity of annotations. In practice, we recommend first running a small-scale pilot study on the task at hand to compare the trade-offs of different LM annotations by defining some simple proxy metrics before performing large-scale annotation and fine-tuning. In general, it is challenging to choose the optimal annotation LM without defining some quantitive measures of the “value for money”, which, as mentioned in our discussion section, is still an open research problem since modeling and optimizing such cost in real-world settings involves many complex factors, e.g., task complexity, desired label quality, and available resources.

---

> > > > ### Comment · Reviewer_ogFx · 2023-08-14
> > > >
> > > > Thanks for your response!

---

### Official Review · Reviewer_YvhV · 2023-07-06

**Soundness:** 4 excellent
**Presentation:** 3 good
**Contribution:** 3 good
**Rating:** 7
**Confidence:** 5

**Summary:**

This paper studies the problem of how to fine-tune a relatively small, domain-specific language model under the constraints of limited annotation resources. The authors propose an Interactive Multi-Fidelity Learning (IMFL) approach. This approach aims to optimize the performance of the fine-tuned language model through multiple rounds of human-model collaborative annotation and fine-tuning. In each round, the method applies an exploration-exploitation query strategy (such as diversity sampling, uncertainty sampling, etc.) to balance high-fidelity human annotations and low-fidelity model annotations. Moreover, the IMFL method also introduces prompt retrieval and variable batch sizes to make better use of the annotated samples. The authors test the proposed IMFL method on four datasets from the financial and medical domains.



**Strengths:**

- In general, I think the experimental design of this paper is sound and the paper is easy to follow and understand.
- I think the problem studied in this paper is interesting and highly meaningful. The authors offer a straightforward and practical strategy for optimizing the performance of the fine-tuned language model under limited human and computational resources. Notably, the designs of the variable batch size and the coordination of human and model annotations in each round are innovative.
- The authors validate their method on four different datasets, consistently outperforming the baseline models in all cases. Particularly, I appreciate the extensive experiments and analyses performed in Section 4 to verify the effectiveness of each component of their framework, along with practical suggestions for future practitioners.

**Weaknesses:**

One major weakness of this paper is its strong assumption that we already have a good pool of unlabeled data, and all the methods are experimented on these pools. However, in many real-world settings, we do not have such an immediately available unlabeled data pool. The quality of the unlabeled data pool might be crucial to the effectiveness of the proposed method. Without relevant experiments, we don't know if the method proposed will still work in real-world settings.


**Questions:**

Why is the budget set only to 1000? While I understand that human annotations may be constrained in real-world applications, I think it would be meaningful to include the performance and analysis of the model under different budget settings in the paper.


**Limitations:**

Yes, the authors have explained the limitations of their study in the paper.

---

> ### Author Rebuttal · Authors · 2023-08-09
>
> Dear reviewer YvhV,
>
> Thank you very much for your detailed and thoughtful comments. We are glad to hear that you found the paper interesting, highly meaningful, and innovative, and our experiments are sound and strong to verify the effectiveness. Below we address the constructive comments and feedback raised in the review.
>
> > **Assumption on the availability and quality of the unlabeled data pool.**
>
> Thanks for the comment. As highlighted in the Discussion and Limitation section, IMFL's performance is limited by the size of the unannotated dataset and the diversity of examples presented in the dataset. This is because IMFL functions by annotating existing samples rather than creating new samples. The authors believe that the assumption of an unlabeled data pool is mild as this is the basis for applying any data-driven / deep learning techniques. Nevertheless, the authors do agree that the quality of the unlabeled data pool is closely related to the final model performance. Therefore, in practice, some basic data wrangling techniques such as data cleaning, down-selection, and filtering, could be employed before applying the proposed IMFL to further improve the final model performance. These efforts are helpful from a practical standpoint but a nuanced study on the data pipeline is beyond the scope of our research.
>
> > **Why is the budget set only to 1000? I think it would be meaningful to include the performance and analysis of the model under different budget settings in the paper.**
>
> Thanks for the valuable suggestion. The authors agree that such analysis is meaningful and thus we have included experiments on the impact of different human annotation budgets and summarized the results in Table 2 in the supplementary material. Our results suggest that reducing the amount of allowed human annotation budget would result in decreased model performance (though still outperforming using all GPT-generated annotations).
>
> In our experiments, we select a default annotation budget of 1000 based on the following considerations: (1) Computational Costs: Using a larger annotation budget would result in an increasing amount of annotation and computational costs for both our experiments and practical deployment. An annotated budget of <= 1000 is a typical setting used by many prior works [2-5] that consider the low-resource setting. (2) Dataset Annotation Constraints: The human annotation budget for our experiments is also limited by the availability of annotated domain-specific datasets. For example, PubMedQA only has 1000 expert-annotated QA instances [1]. In general, the authors believe an annotation budget of 1000 is practically meaningful for NLP tasks in specialized domains like medicine and finance and such experimental design choices should not affect the validity of our findings.
>
> [1] Jin, Qiao, Bhuwan Dhingra, Zhengping Liu, William Cohen, and Xinghua Lu. "PubMedQA: A Dataset for Biomedical Research Question Answering." In Proceedings of the 2019 Conference on Empirical Methods in Natural Language Processing and the 9th International Joint Conference on Natural Language Processing (EMNLP-IJCNLP), pp. 2567-2577. 2019.
>
> [2] Grießhaber, Daniel, Johannes Maucher, and Ngoc Thang Vu. "Fine-tuning BERT for low-resource natural language understanding via active learning." arXiv preprint arXiv:2012.02462 (2020).
>
> [3] Yuan, Michelle, Hsuan-Tien Lin, and Jordan Boyd-Graber. "Cold-start Active Learning through Self-supervised Language Modeling." In Proceedings of the 2020 Conference on Empirical Methods in Natural Language Processing (EMNLP), pp. 7935-7948. 2020.
>
> [4] Schröder, Christopher, Andreas Niekler, and Martin Potthast. "Revisiting Uncertainty-based Query Strategies for Active Learning with Transformers." In Findings of the Association for Computational Linguistics: ACL 2022, pp. 2194-2203. 2022.
>
> [5] Maekawa, Seiji, Dan Zhang, Hannah Kim, Sajjadur Rahman, and Estevam Hruschka. "Low-resource interactive active labeling for fine-tuning language models." In Findings of the Association for Computational Linguistics: EMNLP 2022, pp. 3230-3242. 2022.

---

> > ### Comment · Reviewer_YvhV · 2023-08-17
> >
> > Thank you for the detailed response! I have read the rebuttal, and it has alleviated my concerns raised in the weaknesses.

---

### Official Review · Reviewer_EwLY · 2023-07-07

**Soundness:** 3 good
**Presentation:** 2 fair
**Contribution:** 3 good
**Rating:** 6
**Confidence:** 3

**Summary:**

This article is devoted to a new algorithm for development of small domain-speciﬁc LMs under limited annotation budgets. one of the main ideas of the algorithm is to use as data annotators a combination of a Human and LLM (Large language models). The authors also proposed two innovative developments in their algorithm: 1) prompt retrieval to improve LLM annotation, 2) variable batch size. The paper also presents a new conﬁdence-based query strategy based on applying k-means algorithm to embeddings of the sub-sampled unannotated data and using least conﬁdence for selected items. Of the strengths, I can emphasize the following: a new query strategy for selecting elements is presented, which showed an increase in comparison with other approaches, which introduces scientific novelty to the article. With the help of the tables given in the scientific work, the qualitative increase is shown by the approaches described in the article. Of the weaknesses , I can note that it is not described in detail what happens at the stage “Execute prompt retrieval from Ah” which is described in Algorithm 1. I would like to get a detailed description for "prompt retrieval", for those who are not familiar with this term. There are no tables for hyperparameters with which the models were trained. I would like to see additional statistics on datasets: average, minimum and maximum sentence length.

==== After rebuttal
Thank you for the reply, I have decided raise my score to 6.

**Strengths:**

* The article presents a new algorithm for learning of small domain-speciﬁc LMs under limited annotation budgets the effectiveness of which is confirmed by comparison with various baselines. A new query strategy for selecting elements is presented, the effectiveness of the new strategy is confirmed by experiments. Two innovative designs are proposed to improve the learning process (prompt retrieval to improve LLM annotation, variable batch size), these approaches may be useful in other similar tasks.

**Weaknesses:**

* It is not clear which measure was used in clustering - cosine or euclidean distance, a comparison is needed. There is no information about which model is used to get embeddings in Exploration-Exploitation Query Strategy. Why was Sentence-BERT used to get embedding in Design 1? It seems that you need to use domain-speciﬁc models to get them in both cases. Since we use embedding of proposals, we would like to compare with strategies based on them, for example from the article “Deep Deterministic Uncertainty: A Simple Baseline”. In the article, the authors claim that they selecting cluster centers to reduce intra-iteration redundancy, but after all, at the next iteration, the selected elements in the new subset may be close to those already selected. Why do we reduce redundancy only inside the iteration?
* There are no tables for hyperparameters of models.

**Questions:**

-

**Limitations:**

-

---

> ### Author Rebuttal · Authors · 2023-08-09
>
> Dear reviewer EwLY,
>
> Thank you for your detailed review and suggestions to improve the paper. We are glad to hear that you found that our idea is new, and effective and our designs are innovative and useful in broader tasks.  Below we address the concerns and questions raised in the review.
>
> > **A detailed description of "prompt retrieval" is desired for those who are not familiar with this term.**
>
> Thank you for your suggestion. The goal of prompt retrieval is to retrieve annotated data instances that are semantically similar to the query as in-context learning examples to improve the LLM annotator's performance without the need to fine-tune it. Intuitively, the selected instance-annotation pairs would serve as an illustration to facilitate generating better and more accurate annotations for domain-specific applications. To operationalize this idea, we follow existing studies to retrieve annotated instances that are close to the queried instance in some embedding space that captures semantic or lexical similarities. The design of prompt retrieval is originally described in Section 2.2. Here we provide a more detailed description of the procedure below.
>
> Given the queried instance $x$, annotated data pool $\mathcal{A}$, sentence encoder model (Sentence-BERT), and the number of in-context examples $k$, we execute the following steps:
>
> 1.	Compute the embeddings of the queried instance and instances from the annotated pool.
> 2.	Retrieve the nearest $k$ neighbors of the queried instance $x$, denoted as $x_1, x_2, ...,x_k$, from $\mathcal{A}$ according to a pre-defined similarity metric (e.g., cosine similarity) measured in the embedding space.
> 3.	Concatenate $k$ instances along with their annotations to form the in-context examples, which are then used to construct the prompt for the queried instance to generate annotation from the LLM annotator.
>
> > **Hyperparameters of the model.**
>
> The hyperparameter setting regarding fine-tuning is presented in Table 3 of the supplementary material. We have summarized additional hyperparameters related to prompt retrieval and query strategy in the following table.
>
> | Hyperparameters              | Values                                    |
> |------------------------------|-------------------------------------------|
> | Sentence encoder model       | sentence-transformers/all-mpnet-base-v2   |
> | Dimension of embeddings      | 768                                       |
> | Pooling method               | mean pooling                              |
> | Similarity retrieval metric  | cosine similarity                         |
> | Number of clusters           | annotation budget at a specific iteration |
> | Number of retrieved examples | 5                                         |
> | Size of unlabeled data pool   | 3000                                      |
>
> > **Additional statistics on datasets: average, minimum and maximum sentence length.**
>
> Thanks for the suggestion. We have summarized the additional statistics in the table below, with the average, minimum, and maximum sentence length (in tokens) included, and will add it to the revised version.
>
> | Sentence   length |  FPB  |   Headline   |  MedQA | PubMedQA |
> |:-----------------:|:-----:|:------------:|:------:|:--------:|
> |      Average      | 28.16 |        12.98 | 167.29 |    20    |
> |      Minimum      |   2   |       2      |   17   |     6    |
> |      Maximum      |  148  |      39      |   551  |    50    |
>
> > **The measure used in clustering and the embedding model used in Exploration-Exploitation Query Strategy.**
>
> For k-means clustering, we used Euclidean distance which is the default measure. In the exploration-exploitation query strategy, we use the same Sentence-BERT model as used in prompt retrieval.
>
> > **Why was Sentence-BERT used to get embedding in Design 1? It seems that you need to use domain-speciﬁc models to get them in both cases.**
>
> Our proposed framework is flexible with different embedding models. We choose to use Sentence-BERT in our implementation because it is a general-purpose sentence encoding model which can be employed in different tasks from diverse domains without requiring task-specific fine-tuning. Although encoders fined-tuned on domain-specific tasks would potentially further improve the performance by making the prompt retrieval more effective, it would require more in-domain annotated data, which conflicts with our objective of designing a cost-effective adaptation framework.
>
> > **Why reduce redundancy only inside the iteration?**
>
> The proposed EEQ strategy aims to reduce both intra-iteration and inter-iteration redundancy. Typically, uncertainty-based methods acquire similar samples within an iteration, known as intra-iteration redundancy, while diversity-based approaches acquire similar samples across iterations, known as inter-iteration redundancy. Existing hybrid methods try to avoid intra- and inter-iteration redundancies by combining diversity and uncertainty sampling but may suffer from these redundancies due to unifying the uncertainty and diversity objectives into a single query function, which tends to prioritize one objective over the other.
>
> Our EEQ strategy is an independent two-step selection by first executing the diversity sampling and then the uncertainty sampling. In the first step, we select a subset of an unlabeled data pool consisting of diverse data points in the embedding space. In the second step, we acquire high-uncertainty data points (the ones that are predicted with low confidence by the current model) from the subset.

---

> > ### Author Response · Authors · 2023-08-21
> > **Follow-up on Rebuttal**
> >
> > Dear Reviewer EwLY,
> >
> > We want to thank you for your constructive feedback and thoughtful reviews that helped to improve our paper. As the open discussion deadline is approaching, we would like to take this last opportunity to make sure that all your questions have been properly answered. We would be more than happy to provide more information or clarification should you have any additional questions. If we have addressed your concerns, we would appreciate your consideration in raising the rating to vote toward accepting our paper. Thank you for your continued engagement in advancing our manuscript.
> >
> > Authors

---

### Official Review · Reviewer_Ltph · 2023-07-09

**Soundness:** 3 good
**Presentation:** 2 fair
**Contribution:** 3 good
**Rating:** 6
**Confidence:** 4

**Summary:**

This paper introduces an algorithm to fine-tune LMs for domain specific tasks under a certain budget constraint. They use a mix of human and a high-fidelity LM for annotations. They fix the annotation budget for each and introduce an algorithm to sample from the unannotated pool and distribute it between human labelers and LM annotator. They perform extensive comparisons and ablation studies and show the effectiveness of their approach.

**Strengths:**

- The results section clearly demonstrate the effectiveness of their method under the conditions they considered.
- The paper is well written and easy to follow. Claims are clearly stated, supported by empirical data.
- The results are well explored and discussed. Good intuitions provided.

**Weaknesses:**

- The authors considered very limited tasks. They do not consider the impact of their approach on more generative tasks.
- Their approach and conclusions strongly depends on the LM they used for annotations. Although this is specified in the limitation of their work, more powerful LMs will make the effect of their approach less profound.

**Questions:**

- The authors state "Unfortunately, such an approach is susceptible to the misinformation of LLMs through hallucination, which risks generating unreliable or falsified labels and will, in turn, demolish the model’s utility for high-stakes applications like healthcare and finance, where the truth is of utmost importance." Do they have evidence (from their work or others) that this is the case?
- Are the samples for the GPT-3.5 in Fig 5 drawn randomly or in accordance to the uncertainty score?
- The authors do not discuss how the uncertainty score is calculated. A brief description would be useful.

**Limitations:**

yes

---

> ### Author Rebuttal · Authors · 2023-08-09
>
> Dear reviewer Ltph,
>
> Thank you very much for the constructive comments and feedback. We are happy to hear that the reviewer found our method effective, and our results are well explored and discussed.  Below, we have tried to address all of your feedback and questions. Please take a look and let us know in case you would like additional clarification on any of these points.
>
> > **Impact of approach on more generative tasks.**
>
> Our focus in this work is to build a novel end-to-end framework for cost-effective fine-tuning of LMs on specialized domain-specific tasks that are beyond the scope of standard pre-training. Despite the rarity of such domain-specific benchmarks compared to general NLG tasks, we conducted a relatively comprehensive evaluation of our method, within our allowable computational budget, on four diverse tasks covering two of the most important application domains, i.e., finance and healthcare. We believe our proposed framework can be easily extended to more generative tasks such as open/close-book question-answering provided with proper data annotation and evaluation criteria.
>
> > **More powerful LMs will make the effect of the approach less profound.**
>
> We agree with the reviewer that the performance of our framework is dependent on the LM used for annotation, which is verified in our experiments in Section 4.4. Generally speaking, our method benefits from more powerful LMs that provide higher annotation quality and thus could help to further reduce the cost of human annotations. This does not conflict with our objective. In fact, fine-tuning on domain-specific data is an essential step to obtaining a powerful and reliable LM for highly specialized domains. However, one common challenge faced in practice is the expensive cost of recruiting experienced human annotators in these domains, which our study aims to address. We envision that in real-world development environments, our framework can be applied iteratively on streaming data flows to continuously improve an LM product.
>
> > **Evidence for misinformation of LLMs through hallucination.**
>
> Thanks for the comment. Yes, there are numerous pieces of evidence and experiments in the current literature that support this. For instance,
>
> Dash et al. [1] conducted an evaluation of GPT-3.5 and GPT-4 for supporting real-world information needs in healthcare delivery. Their experiments (see Table 2 in [1]) showed that while general-purpose LLMs are able to provide safe and credible responses, they often do not fully meet the specific information need of a given question, e.g., responses containing hallucinated references.
>
> Karan et al. [2] presented MultiMedQA, a benchmark medical dataset, and evaluated PaLM and Med-PaLM, which perform encouragingly but still reveal key gaps between human evaluation and remain inferior to clinicians. It is observed that models like PaLM and GPT may hallucinate convincing medical misinformation or incorporate biases that could exacerbate health disparities [3].
>
> Bang et al. [4] provided a comprehensive evaluation of ChatGPT using 23 datasets, showing that ChatGPT suffers from hallucination problems, i.e., generated hallucinated information beyond the given knowledge, which is supported by the experiments on misinformation detection related to COVID-19, and factuality on TruthfulQA.
>
> We will add corresponding references in our final version.
>
> [1] Dash, Debadutta, Rahul Thapa, Juan M. Banda, Akshay Swaminathan, Morgan Cheatham, Mehr Kashyap, Nikesh Kotecha et al. "Evaluation of GPT-3.5 and GPT-4 for supporting real-world information needs in healthcare delivery." arXiv preprint arXiv:2304.13714 (2023).
>
> [2] Singhal, Karan, Shekoofeh Azizi, Tao Tu, S. Sara Mahdavi, Jason Wei, Hyung Won Chung, Nathan Scales et al. "Large language models encode clinical knowledge." Nature (2023): 1-9.
>
> [3] Nori, Harsha, Nicholas King, Scott Mayer McKinney, Dean Carignan, and Eric Horvitz. "Capabilities of GPT-4 on medical challenge problems." arXiv preprint arXiv:2303.13375 (2023).
>
> [4] Bang, Yejin, Samuel Cahyawijaya, Nayeon Lee, Wenliang Dai, Dan Su, Bryan Wilie, Holy Lovenia et al. "A multitask, multilingual, multimodal evaluation of ChatGPT on reasoning, hallucination, and interactivity." arXiv preprint arXiv:2302.04023 (2023).
>
> > **Are the samples for the GPT-3.5 in Fig 5 drawn randomly or in accordance with the uncertainty score?**
>
> Thanks for the comment. The samples for the GPT-3.5 in Fig.5 are drawn based on the uncertainty score. We will revise the caption to clarify this.
>
> > **The authors do not discuss how the uncertainty score is calculated. A brief description would be useful.**
>
> Thanks for the suggestion. We've included a brief description of the calculation of the uncertainty score below and will add it to the final version.
>
> For the tasks of natural language generation, the probability of the entire sequence, $\mathbf s$, is the product of the conditional probabilities of new (next) tokens given past tokens, whose resulting log-probability is $\log p( \mathbf s | x) = \sum_{i=1}^{n} \log p(s_i | \mathbf s_{<i})$, where $s_i$ is the $i$-th output token and $\mathbf s_{<i}$ denotes the set of previous tokens. In this work, we define the uncertainty score $C$ by using the geometric mean of the token-level log-probabilities: $C = \frac{1}{n} \sum_{i=1}^n \log p(s_i | \mathbf s_{<i})$. Since the target LM is an offline model with complete white-box access to its token-level log probability, the uncertainty score can be easily calculated during model inference.

---

> > ### Author Response · Authors · 2023-08-21
> > **Follow-up on Rebuttal**
> >
> > Dear Reviewer Ltph,
> >
> > We want to thank you for your constructive feedback and thoughtful reviews that helped to improve our paper. As the open discussion deadline is approaching, we would like to take this last opportunity to make sure that all your questions have been properly answered. We would be more than happy to provide more information or clarification should you have any additional questions. If we have addressed your concerns, we would appreciate your consideration in raising the rating to vote toward accepting our paper. Thank you for your continued engagement in advancing our manuscript.
> >
> > Authors

---

### Author Rebuttal · Authors · 2023-08-09

Dear reviewers,

Thank you all for taking the time to review our paper and we sincerely appreciate all the feedback. In particular, we feel encouraged to see that the reviewers find that

- **The topic of our paper is important, interesting, and practical**:  “This paper is interesting and highly meaningful. The authors offer a straightforward and practical strategy ... ” (reviewer YvhV); "The paper addresses the practical challenges of deploying LLMs in domain-specific tasks" (reviewer ogFx); "This work considers the practical setting of addressing the challenges of deploying LLMs, hence is somehow realistic." (reviewer ogFx).

- **Our idea is innovative**:  “A new query strategy for selecting elements is presented, which introduces scientific novelty to the article” (reviewer EwLY); "Two innovative designs are proposed to improve the learning process (prompt retrieval to improve LLM annotation, variable batch size)" (reviewer EwLY); "Notably, the designs of the variable batch size and the coordination of human and model annotations in each round are innovative" (reviewer YvhV);

- **Our proposed solution is effective**:  "The results section clearly demonstrates the effectiveness of their method." (reviewer Ltph);  "Particularly, I appreciate the extensive experiments and analyses performed in Section 4 to verify the effectiveness of each component of their framework, along with practical suggestions for future practitioners." (reviewer YvhV); "The IMFL framework offers an effective solution for the cost-effective development of small domain-specific LLMs by leveraging multi-fidelity learning." (reviewer ogFx).

- **Our experiments are thorough and the results are promising**:  "The results are well explored and discussed. Good intuitions provided." (reviewer Ltph); "The authors validate their method on four different datasets, consistently outperforming the baseline models in all cases. " (reviewer YvhV); "Experiments on financial and medical tasks provide empirical evidence of the superiority of IMFL over single fidelity annotations." (reviewer ogFx)

- **And our paper is easy to follow and well-written**:  “The experimental design of this paper is sound and the paper is easy to follow and understand.”(reviewer YvhV); “The paper is well written and easy to follow. Claims are clearly stated, supported by empirical data.” (reviewer Ltph).


Based on the reviewer's feedback, we have made the following changes to further improve the clarity of the manuscript:

- Added a brief description of uncertainty score calculation as suggested by reviewer Ltph.

- Added a detailed description of the prompt retrieval and the hyperparameters as suggested by reviewer EwLY.

- Added a statistical analysis of datasets, including the average, minimum and maximum sentence length as suggested by reviewer EwLY.

- Added a discussion and clarification about the exploration-exploitation query strategy, including the details about clustering, embedding model, redundancy, and hyperparameters, as suggested by reviewer EwLY

- Added additional experiments to compare the effect of GPT-3.5 and GPT-4 in terms of annotation quality and performance as suggested by reviewer ogFx

We are committed to address all comments and we welcome any further questions or discussions.

Authors of paper 2221

---

### Comment · Area_Chair_TaMS · 2023-08-12
**Requesting explanations**

Can the authors please explain what is $y_i$ in Equation 4?

---

> ### Author Response · Authors · 2023-08-12
> **Response to AC TaMS - explanations**
>
> Thank you for your question! In Equation 4 we use $y_i$ to denote the prediction made by the model $\theta^{r}$ given input $\mathbf x_i$. To improve clarity and avoid confusion with the ground truth annotation from humans, we will update Equation 4 as follows:
>
> $\mathbf x_i^* = \arg \max_{\mathbf x_i} \left[ 1 - p(f(\mathbf x_i; \theta^{(r)})  | \mathbf x_i; \theta^{(r)}) \right]$

---

### Decision · Program_Chairs · 2023-09-21

**Decision:**

Accept (poster)

**Comment:**

This is a timely paper with sound top-level ideas for blending in human supervision with LLM supervision for training a custom, smaller-scale model.  However, beyond that, the details follow standard practices and not particularly technically deep.  So much more can be done than what the paper proposes.  Hopefully, this paper will trigger follow up work.
Overall all reviewers are positive, and for the strengths mentioned above I recommend accept as a poster.

A minor typo: "closets" should be "closest".